# Continuous Intracranial Pressure Monitoring in Children with ‘Benign’ External Hydrocephalus

**DOI:** 10.3390/jcm14093042

**Published:** 2025-04-28

**Authors:** Maria A. Poca, Diego Lopez-Bermeo, Paola Cano, Federica Maruccia, Carolina Fajardo, Ignacio Delgado, Francisca Munar, Anna Garcia-Merino, Juan Sahuquillo

**Affiliations:** 1Department of Neurosurgery, Vall d’Hebron University Hospital, Passeig Vall d’Hebron 119-129, 08035 Barcelona, Spain; diegofernando.lopez@vallhebron.cat (D.L.-B.); paola.cano@vallhebron.cat (P.C.); anna.garcia@vallhebron.cat (A.G.-M.); sahuquillo@neurotrauma.net (J.S.); 2Neurotraumatology and Neurosurgery Research Unit, Vall d’Hebron Institut de Recerca (VHIR), Vall d’Hebron Hospital Universitari, Vall d’Hebron Barcelona Hospital Campus, Passeig Vall d’Hebron 119-129, 08035 Barcelona, Spain; federica.maruccia88@gmail.com (F.M.); francisca.munar@vallhebron.cat (F.M.); 3Department of Surgery (Neurosurgery), Universitat Autònoma de Barcelona, 08193 Bellaterra, Spain; 4ICFO—Institut de Ciencies Fotoniques, The Barcelona Institute of Science and Technology, 08860 Castelldefels, Spain; carolina.fajardo@icfo.eu; 5Department of Pediatric Neuroradiology, Vall d’Hebron Hospital Universitari, Vall d’Hebron Barcelona Hospital Campus, Passeig Vall d’Hebron 119-129, 08035 Barcelona, Spain; ignacio.delgado@vallhebron.cat; 6Pediatric Anesthesiology Department, Vall d’Hebron Hospital Universitari, Vall d’Hebron Barcelona Hospital Campus, Passeig Vall d’Hebron 119-129, 08035 Barcelona, Spain

**Keywords:** benign enlargement of subarachnoid spaces, external hydrocephalus, intracranial hypertension, intracranial pressure monitoring, intracranial pressure pulsatility, macrocephaly

## Abstract

**Background/Objectives:** This study aimed to evaluate the results of continuous intracranial pressure (ICP) monitoring in children with macrocephaly or rapidly increasing head circumference (HC) diagnosed as benign external hydrocephalus (BEH). Here, we report the absolute ICP measurements, ICP pulsatility, and slow ICP waves after at least 48 h of continuous monitoring in a cohort of 36 children diagnosed with BEH. **Methods:** A prospective study of continuous ICP monitoring was performed in 36 consecutive children with macrocephaly (HC above the 97.5th percentile) or rapidly increasing HC (at least crossing two percentile curves), diagnosed with BEH (22 boys and 14 girls with a mean age of 23.6 ± 13.3 months, minimum: 6, maximum 65), using an epidural sensor. For the first four children in the study, hard copies of the ICP values were obtained using an analog recorder. Starting from the fifth patient, the ICP signal was sampled at 200 Hz and stored on a computer using a computer-based data acquisition and analysis system (LabChart v8.1 software). **Results:** Clinical signs or symptoms were identified in 20 patients (55.6%). Delayed motor or language development was noted in 18 (50%) and 20 (55.6%) patients, respectively. In 13 patients, the enlargement of the subarachnoid spaces was found to be associated with an additional condition. The median of mean ICP values for the entire cohort was 17 mmHg, with a minimum of 6.7 mmHg and a maximum of 29 mmHg. All patients exhibited a percentage of B waves exceeding 20% during the night, with a median value of 47.4% (min: 23.2, max: 75). Three children had nocturnal plateau waves. At night, regular ICP recordings alternated with periods of significant increases in ICP, often exceeding 10 mmHg above baseline values. High-amplitude B waves were noted during these episodes, and the amplitude of the cardiac waveform at the peak of the B waves was consistently greater than 5 mmHg, displaying an abnormal morphology (P2 > P1). A ventriculoperitoneal shunt was implanted in 30 of the 36 patients. **Conclusions:** Patients with BEH may present significant abnormalities in ICP. Monitoring this variable in certain cases can assist in determining the necessity for surgical treatment.

## 1. Introduction

‘Benign’ enlargement of the subarachnoid space (SAS) is one of the causes of macrocephaly in infants. In neuroradiological examinations of children with macrocephaly, this condition is characterized by dilation of the frontal or frontoparietal SAS, typically accompanied by a normal or modest enlargement of ventricular size [1,2]. The terminology used to describe benign external hydrocephalus (BEH) varies widely in the literature, reflecting a lack of consensus and clarity regarding its definition. Terms such as extraventricular hydrocephalus, pseudo-hydrocephalus, benign enlargement of the subarachnoid space, and subdural hygroma have been employed to characterize this condition [3,4]. This diversity in nomenclature underscores the ongoing debate and confusion surrounding the precise nature and diagnostic criteria of BEH [3]. As highlighted in a previous research project conducted by our group [5], we favor the term “benign external hydrocephalus” (BEH), as endorsed by several studies [4,6,7,8]. BEH may be idiopathic (primary) or associated with various genetic and/or acquired conditions (secondary), including mucopolysaccharidoses, achondroplasia, Sotos syndrome, glutaric aciduria type I, or prematurity, among others [1,9], all exhibiting identical neuroradiological findings. In a population-based study conducted in Norway, Wiig et al. reported an incidence of primary BEH in term infants (birth after 37 weeks of gestation) of 0.4 per 1000 live births, with a significant male preponderance [6]. They identified BEH as the most prevalent form of hydrocephalus in young children, accounting for approximately 50% of all pediatric hydrocephalus cases in the region. Despite its high prevalence, BEH remains one of the least studied conditions. This discrepancy may stem from its generally perceived benign clinical course, which could reduce the sense of urgency for further investigation and research [10].

The clinical spectrum of BEH is highly variable, encompassing a range of presentations from asymptomatic macrocephaly to more complex cases involving developmental delays or associated complications. While BEH has traditionally been considered a benign and self-limiting condition, a growing body of evidence suggests a potential association between BEH and an increased risk of subdural hematomas. This risk is thought to arise from the chronic traction exerted on the bridging veins within the dilated SAS [11,12]. Additionally, delays in cognitive and motor development have been reported in children with BEH, with some studies estimating that up to 20% of affected children may experience these developmental delays [6,13]. Other studies have found that while the overall intellectual level of these patients generally falls within the normal range, specific cognitive domains may be affected. When assessed in detail, deficits in attention and perception, as well as subtle impairments in cognitive functioning, have been observed in children with BEH [6,13,14]. Mikkelsen et al. conducted a retrospective study comparing children with BEH to a control population [6]. The children with BEH scored significantly lower in attention, psychomotor speed, executive functions, motor speed and coordination, and verbal intelligence quotient (IQ). These studies cast doubt on the seemingly “benign” nature of BEH, suggesting that the condition may have subtle but significant neurodevelopmental implications.

The presence of abnormalities in cerebrospinal fluid (CSF) dynamics and, in some cases, elevated intracranial pressure (ICP) in children with BEH is suggested by progressive and accelerated skull growth. However, this increase in ICP may go unnoticed while the cranial sutures and fontanelles remain open, as these children do not develop papilledema. The most widely accepted pathophysiological theory to explain BEH involves a delayed or incomplete maturation of the arachnoid granulations. This developmental delay is thought to impair their ability to adequately reabsorb CSF, leading to the accumulation of fluid in the extracerebral spaces [15]. Bateman et al., however, postulate that elevated venous pressure may be the cause of an elevation in CSF pressure, which enlarges the skull relative to brain size while the fontanelles and sutures are open, thus creating a widened SAS [16,17]. Despite these conflicting theories, to the best of our knowledge, ICP has only been monitored in one study—by Shulz et al.—in seven children with BEH, utilizing an external ventricular drain [18]. Four of these patients presented a mean ICP above 10 mmHg and were selected for shunting. However, this study used mean ICP as the primary criterion for shunting and did not assess the frequency of slow ICP waves (A- or B-waves) or evaluate brain pulsatility [18].

Despite being described several decades ago, there is still no consensus on treating patients with BEH versus adopting an expectant management approach, whether in idiopathic or secondary forms. Various therapeutic strategies have been proposed, ranging from observation—which remains the most commonly employed approach—to medical treatment with acetazolamide [19], subdural evacuation punctures [20], or the implantation of CSF shunt systems [20,21]. However, growing evidence highlights the critical importance of early developmental monitoring for children with BEH, combined with timely interventions during key neurodevelopmental stages. We believe that a deeper understanding of the clinical and developmental implications of BEH must be rooted in a more comprehensive analysis of ICP dynamics in these patients. Integrating ICP monitoring into the decision-making process for shunt placement could offer valuable insights, optimize therapeutic strategies, and potentially mitigate the risk of intellectual and motor deficits in affected children. Here, we present the results of continuous ICP monitoring—analyzing absolute ICP measurements, ICP pulsatility, and slow waves—in a cohort of 36 children diagnosed with BEH.

## 2. Patients and Methods

### 2.1. Ethical Statement

The study was approved by the Institutional Ethics Committee of Vall d’Hebron University Hospital (VHUH) under protocol number PR-AMI-459/2017. It was conducted in strict adherence to the ethical principles outlined in the Declaration of Helsinki [22], ensuring the protection of participants’ rights, safety, and well-being. Written informed consent was obtained from the parents or legal guardians of the children prior to their participation in the study.

### 2.2. Patient Selection

A prospective study was conducted from September 2016 to September 2024, enrolling a cohort of 36 consecutive patients diagnosed with BEH at the Pediatric Neurosurgery Unit of Vall d’Hebron University Hospital (VHUH) in Barcelona, Spain. Our hospital is a tertiary care neurosurgery center featuring a Pediatric Neurosurgery Unit, one of the five referral centers in Spain for complex neurosurgical pathology in children. All patients met the clinical criteria for ICP monitoring, and none of the children with BEH underwent shunt placement without prior ICP assessment. The criteria for ICP monitoring in patients diagnosed with BEH at our Pediatric Neurosurgical Unit are the following: 1. Macrocephaly (head circumference [HC] above the 97.5th percentile according to Nellhaus graphs [23]) or a rapidly increasing HC during the first year of life (at least crossing two percentile curves); 2. Enlarged SAS, associated with normal ventricular size or mild ventriculomegaly; and 3. Presence of clinical symptoms or signs, such as irritability, headaches, wide or tense fontanelles, pronounced frontal bossing or dilated scalp veins, among others, and/or persistent psychomotor delay in a six-month follow-up.

Exclusion criteria: Patients were excluded if they had an Evans’ index (EI) > 0.34 or if they were affected by other neurological conditions, including tumors, sequelae of cerebral hemorrhage or infarctions, or any disorder associated with brain atrophy (e.g., perinatal asphyxia, malnutrition, or prior administration of adrenocorticotropic hormone, steroids, or chemotherapeutic agents).

For all children, a comprehensive history was obtained from the parents, including gestational age, birth weight, and details on general motor and cognitive development. Preterm delivery was defined as babies born before 37 weeks of pregnancy. The weight, height, and HC growth charts were evaluated when patients fulfilled the inclusion criteria. Family history and parental HC were registered and classified as macrocephalic if they exceeded the 97.5th percentile of the reference studies for the Spanish population [24]. Since 2017, psychomotor development has been assessed using the Spanish adaptation of the Third Edition of Bayley Scales of Infant and Toddler Development [5,25,26,27].

### 2.3. Neuroimaging Studies

Brain sonography studies are routinely carried out in patients with BEH; if SAS enlargement is accompanied by any degree of increase in ventricular size, HC keeps increasing, or the child has a delay in any psychomotor development milestone, brain Magnetic Resonance Imaging (MRI) is performed. In all patients, at least one cranial sonography and a cranial MRI were performed before ICP monitoring, and the results were independently evaluated by a neuroradiologist (ID). MRI scans were conducted on a Siemens 1.5T Avanto MRI scanner (Erlangen, Germany). Routine unenhanced MRI of the brain includes an isotropic high-resolution T1-weighted 3D image, using a magnetization-prepared rapid acquisition with gradient echo (MPRAGE), axial T2-weighted image (T2WI), axial fluid-attenuated inversion recovery (FLAIR), diffusion-weighted imaging (DWI), susceptibility-weighted imaging (SWI), and 3D Constructive Interference in Steady State (CISS 3D) sequence. The following measurements of the SAS were conducted on the frontal convexities on coronal T1WI or T2WI slices: craniocortical width, sinocortical width, and the anterior part of the interhemispheric fissure [2,28] (Figure 1). The size of the basal cisterns and the Sylvian fissures were arbitrarily classified as normal, reduced, or enlarged. Ventricular size was evaluated by measuring the maximal size of the frontal horns on an axial MRI using the EI [29]. The EI is calculated by dividing the maximum bifrontal distance on an axial MRI slice where the Monro foramina are visible by the maximum inner diameter of the skull at the same level of measurement (Figure 1). Values ≥ 0.30 were considered abnormal [29,30]. The maximum width of the third ventricle was quantified in mm.

### 2.4. Psychomotor Development Assessment

In children between 1 and 42 months of age, the psychomotor development was evaluated using the Third Edition of Bayley Scales of Infant and Toddler Development (Bayley-III) [25,26], with items translated into Spanish [27]. Our previous publication provides a comprehensive overview of the application and methodology of these scales [5]. The first psychomotor evaluation was carried out after the diagnosis. A six-month follow-up was conducted for each child whose results were below the mean in at least one of the five areas of the simple scales and/or in one of the three composite scales. Parents were involved in the evaluation, allowing the therapist to create a more comfortable and spontaneous environment for the babies [5]. In brief, Bayley-III consists of five individual scales: cognitive, receptive, and expressive language, and fine and gross motor scores. Each scale has a simple score that can be combined to produce composite scores: receptive and expressive language to produce a language composite score and fine and gross motor to create a motor composite score. Developmental delay was defined as a scaled score < 7 according to the simple scale [26]. A composite score < 85 was used for the composite scales as the best cut-off measure recommended by Johnson et al. [31].

### 2.5. Methodology for ICP Monitoring

Continuous ICP was performed for at least 48 h in each patient using an extradural sensor (Neurodur-P, Raumedic, Rehau AG + Co, Rehau, Germany) [32]. In the first four children in the study, hard copies of the ICP values were obtained using an analog pen recorder (Yokogawa 3021 Pen Recorder, ADLER S.A., Madrid, Spain) at a paper speed of 20 or 60 cm/h. All other patients were monitored utilizing a custom-designed digital recording technology platform based on the ADInstruments PowerLab 4SP hardware and LabChart v8.1 software (ADInstruments, Ltd., Grove House, Hastings, UK) with the ICP signal sampled at 200 Hz, well above the minimum rate at which digital sampling can accurately record an analog signal (Nyquist Frequency) [33]. ICP monitoring was performed for at least 48 h, including at least two overnight recordings. To reduce signal noise and enhance the detection of slow waves in raw ICP recordings, a second digital channel was created. This channel utilized a smoothing triangular Bartlett digital filter, implemented in the LabChart software with a window length of 255 samples, to produce a time-domain smoothed signal. This filter, similar to the Savitzky-Golay filter used by Riedel et al. [34], reduces the effects of aliasing, and is a good replacement for a traditional moving average [34,35] (Figure 2). For quantitative analysis of the ICP recordings, one of the two senior authors (MAP or JS) reviewed all ICP recordings using LabChart v8.1 and extracted the relevant metrics. Mean ICP: The mean ICP corresponding to the total recording period was calculated manually using the LabChart ‘data pad’. Data obtained from 08:00 to 22:00 were used to calculate the ‘diurnal’ mean ICP. Data obtained from 22:01 to 07:59 were used to calculate the ‘nocturnal’ mean ICP. Definition of slow waves: The presence of slow ICP waves (A- and B-waves) was also quantified according to Lundberg’s original definition [36]. For B-waves, we increased the upper frequency to 3 waves/min [37]. Therefore, B-waves were defined as waves with a frequency of 0.5–3 waves/min, lasting for at least 10 min [36,37,38]. B-waves were subdivided according to amplitude into high-amplitude B-waves (≥10 mmHg) and low-amplitude B-waves (<10 mmHg). The nocturnal rate of pressure waves was calculated by dividing the total duration of pressure waves by the 10-h night sleep period (from 22:01 to 07:59).

ICP pulse amplitude: We used a modification of the method described by Eide et al. [39,40] to calculate the pulsatility of the cardiac wave (ICP_AMP_) in the time domain. ICP_AMP_ is defined as the difference between diastolic minimum pressure and systolic maximum pressure [39]. The amplitude was calculated using LabChart and a 10-min time window in two artifact-free selected periods: (1) when the ICP was stable and at its lowest mean ICP without any A- or B-waves (usually during the day) and (2) when the patient presented a train of B- or A-waves, usually during overnight recordings that have a repetitive pattern during REM (rapid eye movement) sleep [34]. In patients with multiple B- or A-wave trains, we selected the period of time in which the waves had the highest amplitude (Figure 3). The mean ICP of a 10-min time window was reported separately as ICP_AMP1_ (ICP_AMP_ in the regular flat recording) and ICP_AMP2_ (ICP_AMP_ on top of the B- or A-waves). Following the criteria established by Eide et al., an ICP amplitude ≥ 5 mmHg was considered abnormal and indicative of low intracranial compliance [39,41,42]. The methodology for monitoring ICP in children with hydrocephalus is explained in more detail in the Appendix A.

### 2.6. Criteria for Shunting and Postoperative Adverse Events

Due to the limitations of absolute ICP measurements when using extradural sensors, mean ICP was not used as a criterion for determining the need for shunting [43]. Criteria for shunting in these patients were: 1. presence of any frequency of high-amplitude B- or A-waves, and 2. low-amplitude B-waves occurring in more than 10% of the recording time, combined with either ICP_Amp1_ ≥ 5 mmHg or ICP_Amp2_ > 5 mmHg.

The surgical management protocol for CSF shunt placement in these children is detailed in the Appendix A. We defined an adverse event (AE) as any event after surgery that results in an undesirable clinical outcome, prolongs patient hospital stay, requires readmission, a new neurological deficit, requires revision surgery, a new intervention, or contributes to death [44]. Any AE occurring within 30 days post-surgery was classified as early, while those arising after one month were considered delayed.

### 2.7. Statistical Analysis

Statistical analysis was performed using the open-source software Jamovi, version 2.3 (The Jamovi project, 2022. Retrieved from https://www.jamovi.org). The normal distribution of data was tested using the Shapiro–Wilk test. In normally distributed data, the mean, standard deviation (SD), minimum (min), and maximum (max) values were used to summarize the variables. The median, minimum, and maximum values were used in skewed samples. Data were analyzed using X^2^ or Fisher’s exact test to determine differences in categorical variables and the Student’s *t*-test or Mann–Whitney U test to compare continuous variables. All probability values were two-sided, and statistical significance was determined as *p* ≤ 0.05. Data were presented graphically using box-and-whisker plots. Data management: This study was conducted according to the FAIR principles committed to making data and services Findable, Accessible, Interoperable, and Reusable (https://www.go-fair.org). The entire anonymized dataset, metadata, and dictionary are available and can be downloaded from https://zenodo.org (DOI: 10.5281/zenodo.15114835).

## 3. Results

### 3.1. Patient Population and Symptoms

We included 36 children, 22 boys (61.1%) and 14 girls (38.9%), with a mean age of 23.6 ± 13.3 months (min: 6, max: 65). Macrocephaly was observed in 31 children (86.1%). Clinical signs or symptoms other than macrocephaly were identified in 20 patients (55.6%) (Table 1). A positive family history of macrocephaly was present in 8 of the 36 patients (22.2%). Of the 36 patients, 16 (44.4%) had a history of prematurity, with no differences observed between sexes (*p* = 0.593, X^2^ = 0.286). Five of the sixteen premature infants, all born before 32 weeks of gestation, experienced systemic complications during the neonatal period. Delayed motor milestones were observed in 18 patients (50%), while language development impairment was identified in 20 patients (55.6%), based on the Bayley-III test.

In 13 patients, the enlargement of the SAS was found to be associated with an additional intracranial disorder: ectopy of the cerebellar tonsils (*n* = 3), arachnoid cysts (*n* = 3, temporal: 2, posterior fossa: 1), achondroplasia (*n* = 2), macrocephaly-capillary malformation (n = 1), spina bifida (*n* = 1), hydromyelia *(n* = 1), marked plagiocephaly (*n* = 1), and DYRK1A syndrome (*n* = 1). Figure 4 presents various examples of these patients. Table 1 presents additional demographic and clinical characteristics of the patient cohort.

### 3.2. Neuroimaging Data

The measurements of the ventricular system and SAS in the 36 patients studied are detailed in Table 1 and Table 2. The median EI value was 0.29 (min: 0.23, max: 0.34). In 17 of the 36 children studied (47.2%), the EI was <0.30. All patients had frontal CSF compartments (corticocortical, sinocortical, and interhemispheric) > 4 mm. Sylvian fissures and/or basal cisterns were enlarged in 29 children (80.6%). Three children displayed abnormal white-matter (WM) maturation for their age. No statistically significant differences were observed in neuroimaging data when comparing preterm and full-term patients (as detailed in Table 1). Similarly, no significant differences were found between children with isolated BEH and those with BEH associated with additional comorbid conditions (Table 2). These findings suggest that the neuroimaging characteristics of BEH remain consistent across subgroups, reinforcing the idea that BEH exhibits a distinct radiological profile, irrespective of gestational age at birth or the presence of concurrent conditions.

### 3.3. Results of Continuous ICP Monitoring

The results of ICP monitoring were consistent and reproducible. On average, absolute mean ICP values were slightly higher at night than during the day. Frequent artifacts were observed in daytime recordings, whereas nocturnal recordings displayed periods of stability alternating with significant ICP elevations, often exceeding baseline values by more than 10 mmHg. High-amplitude B-waves were frequently observed during these nocturnal episodes. Moreover, the amplitude of the cardiac waveform consistently increased during periods of B-wave activity (Figure 5).

Considering the limitations of absolute values in extradural recordings, the median ICP value for the entire cohort was 17 mmHg (min: 6.7, max: 29 mmHg). There were no significant differences between premature children and those born at term (Table 3, Figure 6). There were no statistically significant differences in mean ICP in nocturnal recordings or in the overall mean ICP between children with primary BEH and those with an associated comorbidity (*p* = 0.212 and *p* = 0.093, respectively) (Table 4, Figure 6). All patients exhibited low- or high-amplitude B-waves during more than 20% of nocturnal ICP recordings, with a median occurrence of 47.4% (min: 23.2, max: 75%) (Figure 6). The median percentage of low-amplitude B-waves was 29.4% (min: 2.5%; max: 58%), while the median for high-amplitude B-waves was 17.9% (min: 0%; max: 50%). High-amplitude B-waves were not detected in only two patients; however, the percentages of low-amplitude B-waves in these patients were 44% and 58%. In three patients, A-waves were detected during nocturnal recordings but were never observed during the day (Table 3 and Table 4).

The amplitude of ICP cardiac waves was analyzed in 31 of the 36 patients using the method described in the Patients and Methods section. The median ICP_AMP1_ was 4 mmHg (min: 2.8; max: 9), and in three of the 31 cases studied, ICP_AMP1_ was >5 mmHg. All patients exhibited a cardiac wave amplitude >5 mmHg during episodes of high-amplitude B-waves or plateau waves, with a median amplitude of 11 mmHg (range: 5.5–21 mmHg). These elevated amplitudes indicated reduced intracranial compliance. Table 3 and Table 4 provide comprehensive data from the ICP recordings, including specific findings in preterm children and those in whom BEH was associated with other comorbidities.

### 3.4. Treatment and Complications

In our study, the ICP sensor required replacement in three of the 36 patients (8.3%) due to accidental dislodgement from the epidural space. Of the 36 families, 31 provided consent for the proposed treatment, leading to the placement of a ventriculoperitoneal CSF shunt in their children. The remaining five families opted to seek a second opinion, which recommended against surgical intervention.

In 11 cases, a Polaris adjustable valve (Polaris Programmable Valve, Sophysa Ltd., Orsay, France) was chosen, initially programmed to 110 mmH_2_O, in-series with a gravity compensating accessory: ShuntAssistant 0/10 (n = 1), 0/20 (n = 7), or 0/25 cmH_2_O (n = 3) (Aesculap-Miethke ShuntAssistant, Christoph Miethke GMBH & Co, Tuttlingen, Germany). The selection of the ShuntAssistant was based on the age and height of each child. In the remaining 20 cases, a Miethke gravitational valve was placed (Aesculap, Tuttlingen, Germany), the most frequently used being the Gav 5/30 cmH_2_O model (n = 16).

Among the 31 children who underwent ventriculoperitoneal CSF shunt placement, the majority (29, 93.5%) experienced an uneventful recovery and were discharged home approximately three days post-surgery. Two patients experienced early postoperative complications: one female patient with a prior history of epilepsy experienced postoperative seizures, and a 10-month-old male patient required surgical revision due to distal catheter malposition. Within the first six months after valve placement, two children developed shunt infections requiring shunt removal. These cases were managed promptly, and both children underwent appropriate antibiotic treatment and subsequent surgical revision as needed. Importantly, no additional valve-related complications, such as mechanical failure, obstruction, or displacement, were observed during the first year after surgery.

Surgical Outcomes: This study aimed to assess ICP abnormalities in children with BEH; therefore, one-year outcomes were not independently evaluated, and developmental delays were not systematically studied. Given this limitation, the neurosurgeon in charge and clinical evaluations reported significant clinical improvements in 90% of the operated cases, while no significant changes were observed in the remaining 10%. Parents frequently reported that children became more active and engaged with their surroundings and showed accelerated language development following shunt implantation. However, these observations were based on clinical impressions and parental reports rather than systematic evaluation, representing a limitation of the study.

## 4. Discussion

The management of BEH remains a topic of ongoing debate, particularly regarding the balance between invasive interventions and conservative monitoring strategies. While some clinicians advocate for early surgical treatment, such as ventriculoperitoneal shunting, to prevent potential neurodevelopmental impairment, others emphasize a more conservative approach, relying on spontaneous resolution and close clinical follow-up. The lack of consensus stems from the variability in clinical presentation, the uncertain long-term consequences of untreated BEH, and the risks associated with surgical procedures. In addition, no well-defined biomarkers exist to predict the natural course of BEH. Such biomarkers are essential for assessing the disorder’s impact, forecasting its progression if left untreated, and determining long-term outcomes.

### 4.1. Pathophysiology of BEH: Unresolved Questions and Controversies

Several risk factors have been associated with the enlargement of the SAS, including prematurity (with or without complications). A hereditary component has also been suggested, with familial occurrence reported in up to 40% of cases of enlarged SAS [3]. However, no definitive underlying cause has been identified, and in most children, the etiology remains idiopathic. In idiopathic cases, two primary pathophysiological theories have been proposed. The first suggests that delayed maturation of arachnoid villi in early infancy leads to impaired CSF absorption, resulting in excessive extracerebral fluid accumulation [1,7,45,46]. Early infancy is a pivotal period for the maturation of the SAS, marked by the development of arachnoid granulations, which play a crucial role in CSF reabsorption. Some authors further suggest that the immaturity of the CSF reabsorption system in children with BEH may be further exacerbated by the concurrent presence of birth-related subdural hemorrhage, potentially contributing to impaired CSF dynamics [45]. Regardless of the etiology, this theory suggests that immature arachnoid granulations in children under two years of age may impair CSF absorption, leading to an imbalance between production and reabsorption. As a result, CSF accumulates in the SAS while the ventricles remain normal or only mildly enlarged.

The second most widely discussed theory regarding the etiology of BEH suggests that dysfunction in the cranial venous drainage system plays a key role. Impaired venous outflow, particularly in the superior sagittal sinus, may lead to altered ICP dynamics and CSF accumulation, contributing to the development of BEH [17,47,48,49]. This theory posits that CSF absorption is directly influenced by the pressure gradient between the SAS and the venous system, making it a critical framework for understanding CSF dynamics and disorders such as BEH. Recent studies [50] and case reports [51] further support this theory, reinforcing the role of impaired venous outflow in the pathophysiology of BEH. MR phlebography was not performed in our patients, however, venous drainage abnormalities may be a contributing factor, particularly in the two cases of BEH associated with achondroplasia, where altered cranial venous dynamics could play a key role in disease development. However, the scarcity of data and the lack of robust animal models leave both the arachnoid-immaturity and venous theories fragmented and incomplete, underscoring the need for further research to fully elucidate the underlying mechanisms of BEH.

### 4.2. Benign External Hydrocephalus: A Misleading Label

BEH is often regarded as a self-limiting condition with no lasting effects; however, growing evidence challenges this assumption [2,7,52]. A previous study conducted by Maruccia et al. revealed that nearly half of the children diagnosed with BEH—excluding cases associated with complicated prematurity, perinatal brain injury, or genetic syndromes—exhibited neurodevelopmental delays in at least one of the Bayley-III scales [5]. This study also found that premature children with BEH had 8.5 times greater odds of presenting any neurodevelopmental delay than full-term children with BEH. This striking disparity underscores the heightened vulnerability of preterm infants to neurodevelopmental challenges, even within the context of BEH [5]. In the present study, clinical signs or symptoms were identified in 20 children (55.6%), while delayed motor development was observed in 18 patients (50%) and delayed language development in 20 patients (55.6%). No statistically significant differences were found between preterm and full-term patients. The high prevalence of motor and language development delays in our study may be influenced by selection bias, as the inclusion criteria for ICP monitoring targeted children with more severe BEH presentations. This approach could have led to an overrepresentation of developmental delays in our cohort.

In long-term follow-up studies, some authors have reported that more than 20% of children with BEH may experience persistent developmental delays, particularly in motor and language domains [7,52,53,54,55,56,57,58]. Yew et al., in a single-center retrospective study, found that 20 out of 99 (21%) patients with BEH not treated surgically exhibited gross motor delays, and four (4%) had a verbal delay at the time of diagnosis [55]. Most of these delays were resolved at a later follow-up of 13 ± 10 months; however, several children had persistent deficits or new verbal deficits [55]. In a separate study, Nasiri et al. conducted a comprehensive developmental assessment of 32 children with BEH at 6, 12, 18, and 24 months of age [58]. They identified developmental delays in 22% of patients, with a higher prevalence among those who presented with macrocephaly at birth or by six months of age [58]. Other studies have shown that BEH children generally achieve an overall intellectual level within the normal range; nevertheless, some may experience persistent challenges in specific cognitive domains or exhibit behavioral disorders [6,13,14,52]. Shen et al. demonstrated that children aged 6 to 24 months with BEH were at a higher risk of developing autism spectrum disorders [56], suggesting a potential association between BEH and neurodevelopmental conditions. Zahl et al. reported reduced quality of life in a long-term follow-up study of a cohort of children with BEH [57].

Despite the limitations of most published studies, including ours, and their retrospective design, growing evidence challenges the perception of BEH as a “benign” condition. The term may be misleading, as many affected children exhibit significant delays in cognitive function, language, motor skills, and social interactions, which can contribute to long-term academic, behavioral, and emotional challenges.

### 4.3. ICP Abnormalities in Children with BEH

The rapid and progressive skull growth observed in children with BEH strongly suggests a significant increase in intracranial volume, which would typically be associated with ICP abnormalities. However, the open sutures and fontanelles in infants allow for cranial expansion, potentially masking the classic symptoms of intracranial hypertension. In children with BEH, ICP disturbances tend to develop gradually in a latent and chronic manner. This capacity for cranial accommodation likely explains why intracranial hypertension is rarely observed in children under three years of age.

Traditionally, even for pediatric neurosurgeons, ICP monitoring has been considered an invasive procedure, often deemed unjustifiable in relatively oligosymptomatic children due to the associated risks. This represents a significant bias, heavily influenced by the general perception implied by the term “benign” in BEH. As a result, there is a notable scarcity of literature on the findings of invasive ICP monitoring in patients with BEH. Several non-invasive methods used to estimate ICP have reported data indicating elevated ICP in children with BEH. Serlin et al. demonstrated anatomical optic nerve (ON) deviations from the norm (wider, longer, and tortuous ON) in 23 infants with an enlarged SAS, with a significant predictive value for neurological complications (at least one witnessed seizure or documented developmental delay) [59].

Studies involving direct CSF pressure measurements in BEH patients have yielded inconsistent results, likely due to variations in methodology, patient selection criteria, and the timing of measurements across different studies. Nogueira and Zaglul reported that none of the cases with BEH in which a lumbar puncture was performed showed evidence of increased ICP [60]. Massager et al. conducted non-invasive ICP monitoring through the anterior fontanelle, demonstrating normal ICP and intracranial compliance in most asymptomatic young children with BEH [61]; however, abnormal ICP recordings were found when children had subdural collections, such as hematomas, hygromas, or progressive hydrocephalus [61]. To our knowledge, the only study in which invasive ICP was monitored was reported by Schulz et al. in 2012 [18]. They monitored ICP using a ventricular catheter for at least two full nights in seven children with BEH. Four out of the seven children without papilledema had a mean ICP > 10 mmHg [18] but the authors did not analyze the presence of slow waves or the ICP_AMP_.

In our cohort, all monitored children exhibited B-waves in more than 20% of their nocturnal recordings, accounting for over 40% of the recording time in 28 cases. B-wave trains usually occurred overnight, during periods likely corresponding to REM sleep, as shown in adults [34]. However, this correlation could not be confirmed due to the absence of simultaneous sleep studies conducted alongside the ICP recordings. During episodes of B-wave trains, the mean ICP increased by more than 10 mmHg above baseline values. Additionally, during either low or high-amplitude B-waves, the amplitude of the cardiac wave (ICP_AMP2_) consistently exceeded 5 mmHg and exhibited a pathological morphology, indicating reduced intracranial compliance. According to Eide et al., in adults, this finding provides important diagnostic information about intracranial compliance, which is valuable for therapeutic decision-making [41,42,62]. Continuous ICP monitoring using epidural sensors is a minimally invasive technique that allows for extended monitoring with a very low complication rate [32]. Despite its primary limitation of an unreliable mean ICP pressure measurement, we consider this method the optimal approach for pediatric patients with BEH. The primary drawback is that absolute ICP values are often artificially elevated. Nonetheless, careful dissection of the epidural space can help minimize discrepancies between ICP values recorded from this compartment and those obtained from other craniospinal regions, improving the accuracy and clinical utility of measurements.

Most of our BEH patients exhibited significant ICP abnormalities and reduced intracranial compliance, indicating that ICP abnormalities may be more central to the pathophysiology of BEH than previously recognized. As suggested by Hanlo et al., it could be hypothesized that chronically elevated ICP, reduced compliance, and abnormal CSF dynamics can reduce cerebral perfusion, potentially harming the brain and impairing myelination during critical stages of development [63]. Incomplete or delayed myelination in the first two years of life can affect motor skills, cognition, and overall neurological development, potentially contributing to developmental delays or neurodevelopmental disorders. Diffusion tensor imaging utilizes the anisotropic diffusion properties of WM to visualize both its structural and functional characteristics within the CNS. Sun et al. studied fractional anisotropy (FA), which measures the degree of directionality of water diffusion. FA increases rapidly during the first few years of life as a result of postnatal myelination, reflecting the progressive maturation and organization of WM tracts. This growth continues until it reaches a plateau around five years of age, marking a critical period for neural connectivity development in the CNS [64]. In a pivotal retrospective study of seventeen children with BEH, Sun et al. found a significant increase in FA and a decrease in mean diffusivity compared with age-matched controls. These differences were particularly pronounced in specific WM regions, including the genu and splenium of the corpus callosum [65], suggesting altered WM microstructure in children with BEH. The authors attributed these findings to increased mechanical pressure on WM tracts, likely due to the accumulation of CSF in the SAS and ventricular system [65]. However, during the follow-up, they observed that these abnormalities gradually improved, returning to normal ranges over time, indicating a potential for WM recovery in BEH-affected children [65].

## 5. Conclusions

In our study, most children who met the inclusion criteria for ICP monitoring exhibited abnormal ICP recordings, characterized by low- and high-amplitude B-waves and, in some cases, A-waves. These findings confirm that chronic intracranial hypertension occurs in some BEH patients during a critical period of brain development, coinciding with active myelination and the formation of neural connectivity in the CNS. ICP abnormalities may interfere with skill acquisition and potentially lead to permanent developmental deficits. Our findings suggest that children with BEH and clinical symptoms—particularly those with persistent language or motor development delays—require close clinical follow-up to monitor their neurodevelopmental progress. In selected children, continuous extradural ICP monitoring is a safe, low-risk procedure that provides valuable insights into ICP dynamics and intracranial compliance. This information aids pediatric neurosurgeons in guiding families toward the most appropriate treatment options, ensuring timely interventions when necessary.

## 6. Limitations of the Study

This study has several limitations. One key limitation is the selection bias introduced by our inclusion criteria, as our center has historically utilized ICP monitoring in the management of both adult and pediatric hydrocephalus. This may have led to an overrepresentation of patients with more severe or atypical cases, potentially limiting the generalizability of our findings to broader BEH populations. A second limitation is our use of epidural ICP monitoring, which may compromise the accuracy of absolute mean ICP values [43]. However, as demonstrated in our previous study [43] and supported by other researchers, epidural ICP readings can be reliably superimposed on those obtained from sensors placed in other craniospinal compartments [66,67,68,69]. We consider epidural ICP monitoring a valid alternative for assessing ICP dynamics in pediatric patients. Previous studies have shown that ICP amplitude and the frequency of slow waves are not biased when compared to intraparenchymal and intraventricular sensors [70]. This suggests that, despite potential discrepancies in absolute ICP measurements, overall ICP dynamics, ICP pulsatility, and waveform analysis are reliable and comparable to more invasive monitoring techniques [43,70].

To assess developmental milestones, we used the Third Edition of the Bayley Scales of Infant and Toddler Development (Bayley-III), one of the most widely recognized and frequently used measures of pediatric development, included in numerous Spanish studies. However, a key limitation is that Bayley-III was standardized on an American pediatric population, and recent research has raised concerns about its validity and accuracy in different geographic and cultural contexts [71]; therefore, this potentially limits the generalizability of our findings.

## 7. Future Work

A common question is whether conducting invasive studies on children with BEH is justified, given that it has traditionally been considered a self-limiting condition. However, growing evidence suggests that BEH may not always be as benign as previously thought, with some children experiencing persistent developmental delays, cognitive challenges, and behavioral issues. Here, we show that most children exhibit significant to severe ICP abnormalities and reduced intracranial compliance, despite having an open cranium. Abnormal ICP is a widely used surrogate biomarker in both adult and pediatric hydrocephalus to assess the need for shunt placement, yet it has not been routinely applied in BEH. A common perception is that ICP monitoring in children is too risky and provides limited clinical benefits. However, our data challenge this view. Revaluating the role of ICP monitoring could be essential in identifying children who may benefit from early intervention and a more individualized management approach. Selecting a minimally invasive yet reliable method for ICP monitoring, such as epidural ICP monitoring, offers significant advantages in children with BEH. By preserving the dural covering intact, this approach reduces the risk of hemorrhagic complications and lowers the likelihood of infection.

Future studies on BEH should include a comprehensive evaluation of developmental outcomes in children along with long-term follow-up using standardized scales. Evaluating cognitive, motor, and behavioral development over time in a large cohort of children is crucial for gaining deeper insights into the long-term impact of BEH. This gap in research still limits our understanding of the true nature of the condition, including the potential for transient or sustained increases in ICP and their impact on neurodevelopment. Addressing this bias and expanding research efforts, particularly through invasive ICP studies, are essential for advancing our understanding of BEH pathophysiology. Such investigations will facilitate the development of more personalized and evidence-based management strategies, ultimately improving outcomes for affected children.

## Figures and Tables

**Figure 1 jcm-14-03042-f001:**
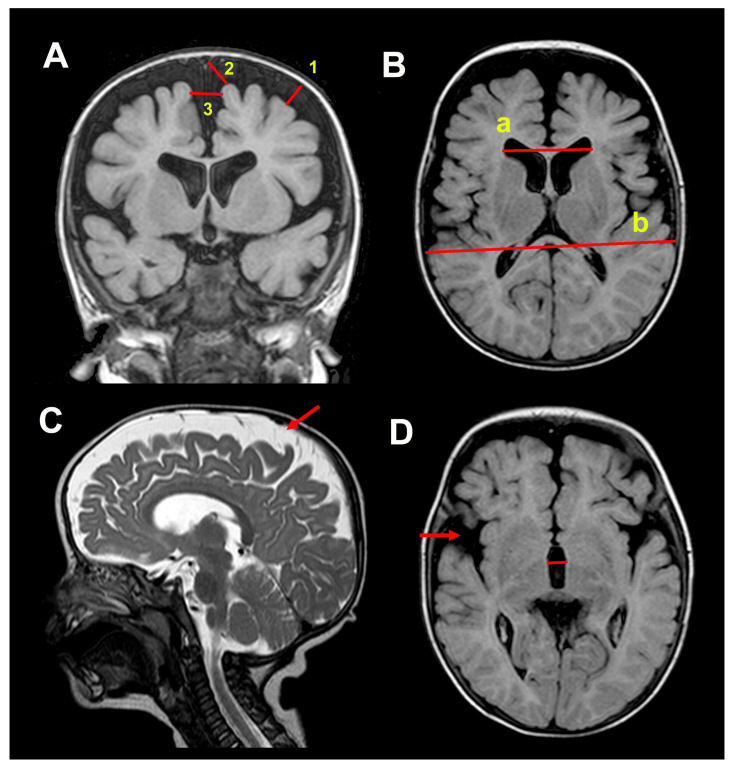
(**A**). Measurements of the frontal subarachnoid spaces: craniocortical width (1), sinocortical width (2), and interhemispheric distance (3). (**B**). Ventricular size evaluation using the Evans’ index (EI) calculated in the axial plane at the level of the foramen of Monro: EI = a/b: maximum bifrontal distance in the axial MRI slice by the maximum inner diameter of the skull at the same level of measurement; in this patient, the EI was 0.34. (**C**). Sagittal image showing the cerebrospinal fluid surrounding the cerebral cortex (red arrow). (**D**). Maximum width of the third ventricle (red line). The image shows an increase in sylvian fissures (red arrow).

**Figure 2 jcm-14-03042-f002:**
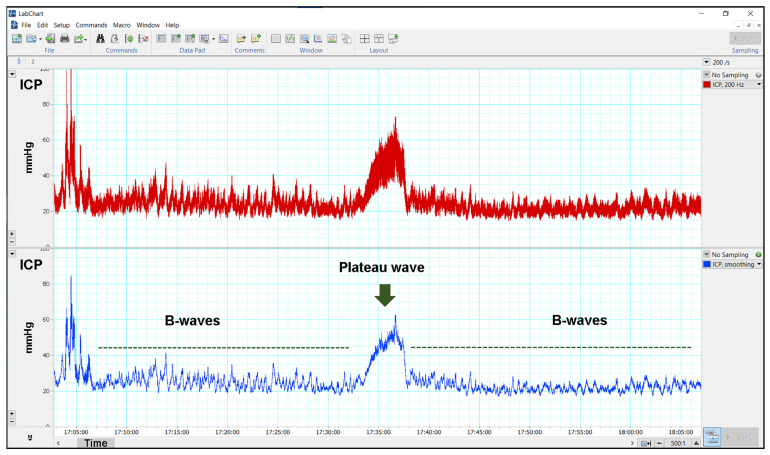
Continuous intracranial pressure (ICP) was recorded using an epidural sensor (Neurodur-P, Raumedic, Rehau AG + Co, Rehau, Germany), along with PowerLab 4SP hardware and LabChart 8.1 software for analysis. In the (**top**) section, the raw ICP signal is displayed, sampled at 200 Hz (in red). The (**bottom**) section shows the same ICP signal as the upper channel but visualized through a smoothing triangular Bartlett digital filter, implemented in LabChart software, which generated a time-domain smoothed signal (in blue). Note how the B waves (green dashed lines) are much clearer in the bottom ICP recording.

**Figure 3 jcm-14-03042-f003:**
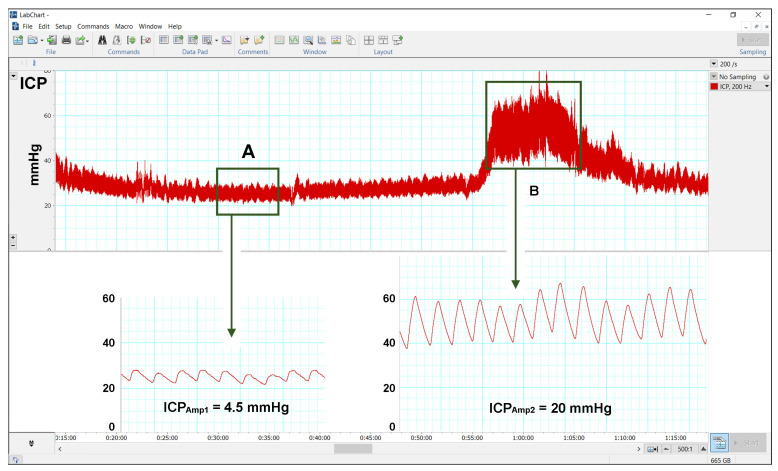
Pulsatility of the cardiac wave in the time domain. The amplitude of ICP cardiac wave changes when analyzed in a segment of the regular ICP recording (**A**) and at the top of B waves or during plateau waves (**B**). In this girl, the amplitude increases from 4.5 mmHg (ICP_AMP1_) to 20 mmHg (ICP_AMP2_).

**Figure 4 jcm-14-03042-f004:**
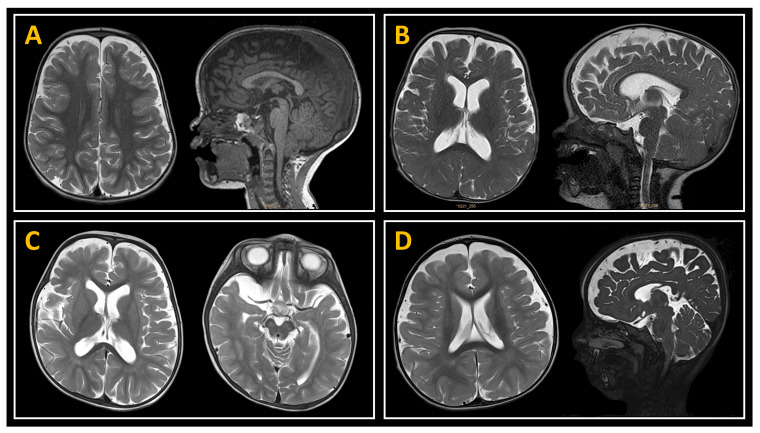
Examples of children with benign external hydrocephalus related to other conditions: (**A**) Ectopia of the cerebellar tonsils; (**B**) Achondroplasia; (**C**) Right temporal arachnoid cyst with ipsilateral occipital plagiocephaly; (**D**) Bilateral occipital plagiocephaly. All four children exhibited ongoing symptoms and/or delays in their motor or language development.

**Figure 5 jcm-14-03042-f005:**
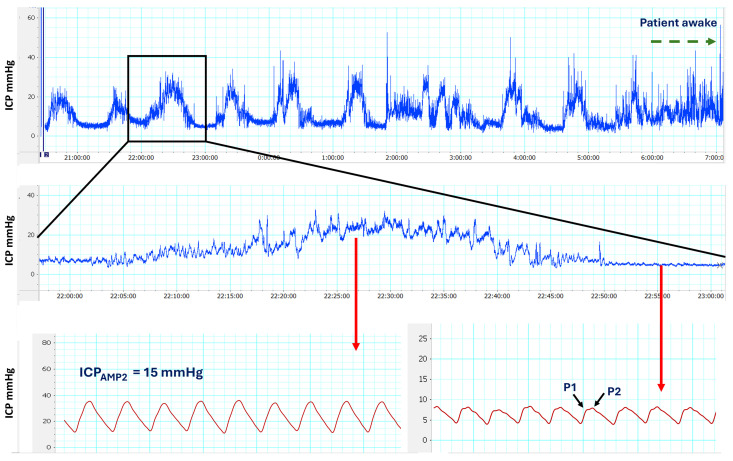
Continuous intracranial pressure (ICP) recording in a seven-month-old girl with BEH with a head circumference of 48.5 cm (percentile > 99th). (**Top**): ICP recording of an entire night—between 20:40 to 07:00 h—that shows a pattern where regular recording periods alternate with pathological periods with marked increases in ICP values that exceed 20 mmHg with respect to baseline values. At the end of the recording, the girl wakes up, and motion artifacts appear (green dashed arrow). In the middle section (**middle**), there is an expansion of a segment from the ICP recording in part A (highlighted within the black square). Here, we can observe slow ICP waves and an increase in mean ICP values. Bottom: we provide details of the ICP cardiac wave at the peak of a B wave (ICP_AMP2_) (red arrow), which exhibits an amplitude of 15 mmHg (**bottom-left**), alongside the ICP cardiac wave during a regular recording segment (ICP_AMP1_) (red arrow), which has an amplitude of 3 mmHg (**bottom-right**). In D, the P2 component is higher than the P1 component.

**Figure 6 jcm-14-03042-f006:**
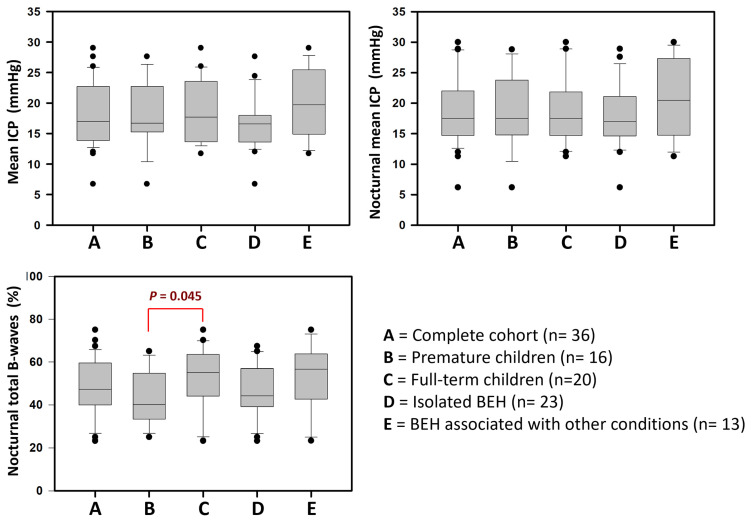
Box-and-whisker plots showing: (**Top left**): mean intracranial pressure (ICP); (**Top right**): nocturnal mean ICP; (**Bottom left**): total percentage of nocturnal B-waves. In each plot, we can observe data from the entire cohort (A) and the different subgroups of patients: preterm (B), full-term (C), isolated benign external hydrocephalus (BEH) (D), and BEH associated with other conditions (E).

**Table 1 jcm-14-03042-t001:** Demographic, clinical, and neuroimaging data from the total cohort of monitored patients and the subgroups of pre-term and full-term patients.

	Pre-Term	Full-Term	Total Cohort	*p* *
** *n* **	16	20	36	
**Birth events**				
Sex (boys/girls)	9/7	13/7	22/14	0.593
Gestational age (weeks)	32.5 (25–36)	39 (37–41)	37 (25–41)	**<0.001**
Apgar score 5′	9 (3–10)	9 (6–10)	9 (3–10)	0.137
Apgar score 10′	9.5 (5–10)	10 (9–10)	10 (5 –10)	**0.028**
Birth weight in kg	2 (0.57–3.04)	3.3 (2–4.5)	2.65 (0.57–4.5)	**<0.001**
Head circumference (cm)	32.5 (24.5–41)	35.5 (32–38)	34.3 (24.5–41)	**0.007**
Abnormal neonatal period ** (n)	5	0	5	
Associated conditions (n)	4	6	10	
**Findings at the time of ICP monitoring**				
Age (months, mean ± SD)	23.9 ± 9.34	23.4 ± 16.1	23.6 ± 13.3	0.524
Macrocephaly (yes)	12	19	31	0.149
Presence of clinical symptoms or signs (yes)	8	12	20	0.549
- Irritability	0	3	3	
- Hypotonia	7	5	12	
- Headaches	1	2	3	
- Wide anterior fontanel	0	2	2	
- Dilated scalp veins	0	2	2	
- Delayed motor development ***	9	9	18	0.502
Delayed language development ***	9	11	20	0.922
Positive family history of macrocephaly (yes)	2	6	8	0.257
**Neuroimaging data**				
Evans’ index	0.29 (0.27–0.32)	0.3 (0.23–0.34)	0.29 (0.23–0.34)	0.654
Third ventricle (mm)	7.8 (3.2–11)	9.8 (3–14)	8.3 (3–14)	0.130
Craniocortical width (mm)	10.7 (7–18)	10.9 (7.1–17)	10.8 (7–18)	0.949
Sinocortical width (mm)	12 (4.6–20.7)	11 (5–15.8)	11.5 (4.6–20.7)	0.166
Interhemispheric fissure (mm)	12.2 (6–24.7)	10.5 (6.3–22.5)	11.4 (6–24.7)	0.535
Enlarged Sylvian fissures (yes)	12	13	25	0.718
Enlarged basal cisterns (yes)	10	15	25	0.483
Abnormal white-matter maturation (yes)	2	1	3	0.574

Results are expressed as median (minimum-maximum). ICP: Intracranial pressure; SD: Standard deviation. * *p*-values when comparing data obtained from preterm and full-term patients (Mann–Whitney U test) ** Systemic neonatal complications. *** Evaluated using the Third Edition of Bayley Scales of Infant and Toddler Development (Bayley-III) [25,26]. Significant differences between the two groups are highlighted in bold.

**Table 2 jcm-14-03042-t002:** Demographic, clinical, and neuroimaging data for the total cohort and subgroups of children with isolated BEH, and those with BEH associated with other conditions.

	Isolated BEH	BEH Associated with Other Conditions	Total Cohort	*p* *
** *n* **	23	13	36	
**Birth events**				
Sex (boys/girls)	13/10	9/4	22/14	0.501
Gestational age (weeks)	36 (29–40)	38 (25–41)	37 (25–41)	0.136
Apgar score 5′	9 (3–10)	9 (6–10)	9 (3–10)	0.729
Apgar score 10′	10 (5–10)	10 (9–10)	10 (5 –10)	0.674
Birth weight in kg	2.31 (0.57–4.5)	3.3 (1.04–3.8)	2.65 (0.57–4.5)	**0.043**
Head circumference (cm)	34 (24.5–41)	36 (27–38)	34.3 (24.5–41)	0.092
**Findings at the time of ICP monitoring**				
Age (months)	22 (6–65)	19 (8–33)	22 (6–65)	0.269
Macrocephaly (yes)	20	11	31	1
Presence of clinical symptoms or signs (yes)	8	12	20	**0.001**
- Irritability	0	3	3	
- Hypotonia	6	6	12	
- Headaches	1	2	3	
- Wide anterior fontanel	1	1	2	
- Dilated scalp veins	2	0	2	
Delayed motor development **	10	8	18	0.489
Delayed language development **	13	7	20	1
Positive family history of macrocephaly (yes)	3	5	8	0.107
**Neuroimaging data**				
Evans’ index	0.30 (0.24–0.34)	0.26 (0.23–0.34)	0.29 (0.23–0.34)	0.097
Third ventricle (mm)	8.2 (3.2–14)	8.4 (3–12.7)	8.3 (3–14)	0.767
Craniocortical width (mm)	10.5 (7–18)	12 (7.1–17)	10.8 (7–18)	0.468
Sinocortical width (mm)	12 (4.6–20.7)	11.5 (5–14)	11.5 (4.6–20.7)	0.365
Interhemispheric fissure (mm)	11 (6–24.7)	13 (6.3–16.5)	11.4 (6–24.7)	0.315
Enlarged Sylvian fissures (yes)	15	10	25	0.708
Enlarged basal cisterns (yes)	16	9	25	1
Abnormal white-matter maturation (yes)	3	0	3	0.288

Results are expressed as median (minimum-maximum). BEH: Benign external hydrocephalus; ICP: Intracranial pressure; SD: Standard deviation. * *p*-values when comparing data obtained from patients with isolated BEH and patients with BEH associated with other conditions (Mann–Whitney U test). ** Evaluated using the Third Edition of Bayley Scales of Infant and Toddler Development (Bayley-III) [25,26]. Significant differences between the two groups are highlighted in bold.

**Table 3 jcm-14-03042-t003:** Intracranial pressure data for the total cohort and the subgroups of pre-term and full-term patients.

	Pre-Term	Full-Term	Total Cohort	*p* *
** *n* **	16	20	36	
Mean ICP (mmHg)	16.7 (6.7–27.6)	17.7 (11.7–29)	17 (6.7–29)	0.811
Diurnal mean ICP (mmHg)	17.7 (7–27)	16.7 (9.5–31)	17.6 (7–31)	0.828
Nocturnal mean ICP (mmHg)	17.5 (6.2–28.8)	17.5 (11.3–30)	17.5 (6.2–30)	0.973
**Slow ICP waves** (%)				
Nocturnal low-amplitude B-waves	26.7 (2.5–43)	34.9 (6.5–58)	29.4 (2.5–58)	**0.042**
Nocturnal high-amplitude B-waves	17.9 (8.6–37.6)	18.4 (0–50)	17.9 (0–50)	0.811
Nocturnal total B-waves	40 (25–65)	55 (23.2–75)	47.4 (23.2–75)	**0.045**
Presence of plateau waves (yes)	1	2	3	
**Cardiac ICP waves**				
** *n* **	14	17	31	
**Amplitude**				
Regular ICP recording (mmHg)	4 (3–9)	4 (2.8–5.5)	4 (2.8–9)	0.516
Top of the B-waves (mmHg)	12.5 (8–21)	10 (5.5–18)	11 (5.5–21)	**0.033**
**Morphology of cardiac ICP waves**				
Abnormal morphology in regular recording (yes)	5	2	8	
Abnormal morphology at the top of the B-waves (yes)	14	17	31	

Results are expressed as median (minimum-maximum). ICP: Intracranial pressure. * *p*-values when comparing data obtained from preterm and full-term patients (Mann–Whitney U test). Significant differences between the two groups are highlighted in bold.

**Table 4 jcm-14-03042-t004:** Intracranial pressure data for the total cohort and the subgroups of children with isolated BEH and those with BEH associated with other conditions.

	Isolated BEH	BEH Associated with Other Conditions	Total Cohort	*p* *
** *n* **	23	13	36	
Mean ICP (mmHg)	16.6 (6.7–27.6)	19.7 (11.7–29)	17 (6.7–29)	0.093
Diurnal mean ICP (mmHg)	14.2 (7–25.8)	22.3 (12.2–31)	17.6 (7–31)	**0.013**
Nocturnal mean ICP (mmHg)	17 (6.2–28.9)	20.5 (11.3–30)	17.5 (6.2–30)	0.212
**Slow ICP waves** (%)				
Nocturnal low-amplitude B-waves	29.4 (2.5–44.2)	27.4 (7.5–58)	29.4 (2.5–58)	0.645
Nocturnal high-amplitude B-waves	16.7 (6.7–38)	20 (0–50)	17.9 (0–50)	0.458
Nocturnal total B-waves	44.2 (23.2–67.4)	56.7 (23.3–75)	47.4 (23.2–75)	0.166
Presence of plateau waves (yes)	2	1	3	
**Cardiac ICP waves**				
** *n* **	20	11	31	
**Amplitude**				
Regular ICP recording (mmHg)	4 (2.8–5)	4 (3–9)	4 (2.8–9)	0.169
Top of the B-waves (mmHg)	11 (5.5–16)	11 (6–21)	11 (5.5–21)	0.468
**Morphology of cardiac ICP waves**				
Abnormal morphology in regular recording (yes)	4	4	8	
Abnormal morphology at the top of the B-waves (yes)	20	11	31	

Results are expressed as median (minimum-maximum). ICP: Intracranial pressure. * *p*-values when comparing data obtained from patients with isolated BEH and patients with BEH associated with other conditions (Mann–Whitney U test). Significant differences between the two groups are highlighted in bold.

## Data Availability

Data management: This study was conducted according to the FAIR principles committed to making data and services Findable, Accessible, Interoperable, and Reusable (https://www.go-fair.org). The entire anonymized dataset, metadata, and dictionary are available and can be downloaded from https://zenodo.org (DOI: 10.5281/zenodo.15114835).

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
