# Peer review of "Continuous Intracranial Pressure Monitoring in Children with ‘Benign’ External Hydrocephalus"

_jcm, 2025, doi:10.3390/jcm14093042_

Round 1

Reviewer 1 Report

Comments and Suggestions for Authors

This is a well-written paper that clarifies a field where ICP monitoring is not traditionally done, given the associated risks in young infants. The design is well done, and the figures are teaching and illustrative for beginners. The evidence is well supported, and it could imply a change of paradigm in the way patients with BEH are managed, using ICP continuous monitoring to detect those patients who could benefit from early surgical treatment.

Author Response

Dear Editors,

We appreciate the reviewers' time and comments regarding our article, to which we respond individually.

Reviewer 1

Reviewer 1: “This is a well-written paper that clarifies a field where ICP monitoring is not traditionally done, given the associated risks in young infants. The design is well done, and the figures are teaching and illustrative for beginners. The evidence is well supported, and it could imply a change of paradigm in the way patients with BEH are managed, using ICP continuous monitoring to detect those patients who could benefit from early surgical treatment.”

Our answer: We sincerely thank Reviewer 1 for their insightful and constructive comments on our manuscript.

Reviewer 2 Report

Comments and Suggestions for Authors

The authors present their results of systematically monitoring ICP in a cohort of children with enlarges/increasing head circumference (HC) and a clinical diagnosis of BEH. They describe the varying and often subtle clinical picture, which is often missed, the lack of consensus and doubts regarding this condition. They support this introduction with a reference list including the most relevant publications.

Based on this, their proposal to conduct >48 hr. ICP monitoring is very relevant aiming directly at a better understanding of BEH which is a prerequisite for optimal management.

The cohort is described with consecutive inclusion, so it can be assumed that inclusion was performed unbiased and systematically. The inclusion criteria based on HC, imaging and symptoms are clearly described.

The paragraph “Methodology for ICP monitoring” is very long and could be considerably abbreviated without loss of important information – in can safely be assumed that the neurosurgical reader is already acquainted with the basic principles of ICP monitoring, and this background information can thus be covered by referring to the relevant publications in the reference list.

The description of the protocol for shunt placement and outcome following this is unnecessarily detailed – the paper deals with the BEH diagnosis and indication for and not with shunt surgery per se. Additionally the text in 3.4. Treatment and complications could be limited to the last paragraph only – this is almost entirely a repetition of paragraph 2.7.

Figure 4 seems redundant – the diagnostic principles are already clearly shown in figure 1.

Figures 2, 3, 6 and 7 contain overlapping information – I think that two figures would be sufficient to support the text.

It would also be helpful to make the discussion shorter and sharper, as much of the very relevant results drown in speculations.

In summary, the paper is very relevant, but much too long, as it contains lengthy text sections with basic background information, irrelevant text on the local shunt protocol and repeated information. My feeling is that the legibility would increase considerably for most pediatric neurosurgeons if the text would be slimmed by at least half of the current extent.

Author Response

Dear Editors,

We appreciate the Reviewer's time and comments regarding our article. Below you will find our replies to all the Reviewer’s remarks and a description of the changes we have made to our manuscript. The modifications have been marked in red (deleted) or blue (changed or updated text). We hope Reviewer 2 approves of our changes.

Reviewer 2

Reviewer 2: The authors present their results of systematically monitoring ICP in a cohort of children with enlarges/increasing head circumference (HC) and a clinical diagnosis of BEH. They describe the varying and often subtle clinical picture, which is often missed, the lack of consensus and doubts regarding this condition. They support this introduction with a reference list including the most relevant publications. Based on this, their proposal to conduct >48 hr. ICP monitoring is very relevant aiming directly at a better understanding of BEH which is a prerequisite for optimal management. The cohort is described with consecutive inclusion, so it can be assumed that inclusion was performed unbiased and systematically. The inclusion criteria based on HC, imaging and symptoms are clearly described.

Our answer: We sincerely thank Reviewer 2 for their insightful and constructive comments on our manuscript.

Reviewer 2:  The paragraph “Methodology for ICP monitoring” is very long and could be considerably abbreviated without loss of important information – in can safely be  assumed that the neurosurgical reader is already acquainted with the basic principles of ICP monitoring, and this background information can thus be covered by referring to the relevant publications in the reference list.

Our answer: In this section, we would like to respectfully note a divergence of opinion with the reviewer regarding the commonly accepted indications for intracranial pressure (ICP) monitoring in pediatric patients with hydrocephalus, as numerous studies have shown that this methodology is not routinely employed in many pediatric neurosurgical centers. Nevertheless, we recognize the reviewer’s viewpoint and, accordingly, have significantly abbreviated this section in the main text, transferring the detailed discussion to the Supplementary Material. This allows interested readers to consult the detailed methodology of ICP monitoring in pediatric patients, while enabling others who are already acquainted with the procedure to focus on the main text. Additionally, we are unable to cite any prior publication from our group that presents the level of detail included in this section. We hope that this revision will be acceptable to Reviewer 2.

In the revised version of the article and in this specific section, we have removed the following text of the original article:

  • Lines 257 to 266
  • Lines 272 to 277
  • Lines 279 to 280
  • Lines 283 to 285
  • Lines 297 to 300
  • Lines 311 to 314

Reviewer 2: The description of the protocol for shunt placement and outcome following this is unnecessarily detailed – the paper deals with the BEH diagnosis and indication for and not with shunt surgery per se. Additionally the text in 3.4. Treatment and complications could be limited to the last paragraph only – this is almost entirely a repetition of paragraph 2.7.

Our answer: In response to Reviewer 2’s suggestions, we have removed the detailed protocol for cerebrospinal fluid (CSF) shunt placement from Section 2.6 of the Patients and Methods (lines 331 to 350 of the original article) and have instead transferred this content to the Supplementary Materials. We believe that the low complication rate associated with shunt placement in pediatric patients at our center is, to a great extent, attributable to the systematic implementation of this protocol.

To consolidate all surgical information and improve clarity, we have implemented the following additional revisions:

  • In accordance with the reviewer’s suggestions, we have eliminated redundant content between Sections 2.6 and 3.4 (lines 327 to 331 in the original article), thereby streamlining the presentation of the surgical methodology.
  • Sections 2.6 and 2.7 have been merged into a single section entitled 2.6. Criteria for Shunting and Postoperative Adverse Events. Furthermore, content from lines 352 to 361 has been removed, with selected portions now included in the Supplementary Materials.
  • A portion of the text on lines 447 and 448 in Section 3.4 has also been removed to eliminate repetition and improve clarity.

These modifications were made to improve the manuscript’s structure, minimize redundancy, and provide a clearer, more streamlined presentation of the surgical aspects of our study. We hope that these modifications are in line with Reviewer 2’s recommendations.

Reviewer 2:  Figure 4 seems redundant – the diagnostic principles are already clearly shown in figure 1.

Our answer: Figure 4 of the original article has been removed

Reviewer 2:  Figures 2, 3, 6 and 7 contain overlapping information – I think that two figures would be sufficient to support the text.

Our answer: Our paper examines the findings related to intracranial pressure in children diagnosed with benign external hydrocephalus. We believe that the information presented in Figures 2, 3, and 7 contributes meaningfully to the reader’s understanding of the methodology employed and underscores the significant intracranial pressure abnormalities that may be observed in these patients. However, in accordance with the reviewer’s recommendation, we have removed Figure 6 from the original version of the manuscript.

Reviewer 2:  It would also be helpful to make the discussion shorter and sharper, as much of the very relevant results drown in speculations.

Our answer: In response to the reviewer's suggestions, we have significantly shortened the overall discussion. However, we have retained the segment on ICP findings in children with BEH, as this is the main focus of the article. Additionally, we have moved the section discussing the pathophysiology of BEH to the beginning of the discussion section, following a suggestion from Reviewer 3.

Reviewer 2:  In summary, the paper is very relevant, but much too long, as it contains lengthy text sections with basic background information, irrelevant text on the local shunt protocol and repeated information. My feeling is that the legibility would increase considerably for most pediatric neurosurgeons if the text would be slimmed by at least half of the current extent.

Our answer: We are sincerely grateful for the reviewer’s valuable input and believe that the revisions made in response to the comments have strengthened the manuscript. We hope that the revised version will be considered suitable for publication in the Journal of Clinical Medicine, Special Issue: State of the Art in Pediatric Neurosurgery.

Reviewer 3 Report

Comments and Suggestions for Authors

The study shows that patients with benign external hydrocephalus may have high intracranial pressure, suggesting that monitoring ICP could be crucial in decision for surgical treatment.Based on the study, here are some suggestions.

  1. If the study can analyze the correlation between ICP parameters (e.g., mean ICP, B-wave frequency) and neuroimaging data, it will be batter.
  2. In the Figure 4. (A), please make sure that c is preoperative MRI or postoperative MRI?
  3. This paragraph of Pathophysiology of BEH could be shortened and placed at the beginning of the discussion section.

Author Response

Dear Editors,

We appreciate the Reviewer's time and comments regarding our article. Below you will find our replies to all the Reviewer’s remarks and a description of the changes we have made to our manuscript. The modifications have been marked in red (deleted) or blue (changed or updated text). We hope Reviewer 3 approves of our changes.

Reviewer 3

The study shows that patients with benign external hydrocephalus may have high intracranial pressure, suggesting that monitoring ICP could be crucial in decision for surgical treatment. Based on the study, here are some suggestions.

Reviewer 3: If the study can analyze the correlation between ICP parameters (e.g., mean ICP, B-wave frequency) and neuroimaging data, it will be better.

Our answer: We agree that the analysis suggested by Reviewer 3 would undoubtedly offer a more robust and comprehensive understanding of the data. However, this was a prospective pilot study based on continuous intracranial pressure (ICP) monitoring conducted at a single center, involving 36 consecutive children presenting with macrocephaly. Due to the limited sample size, performing a correlation analysis between ICP parameters and specific neuroradiological findings—although undoubtedly of great interest—would result in small subgroups. This would likely introduce statistical artifacts and reduce the reliability of the results, potentially raising concerns among reviewers regarding the validity of such analyses. Nevertheless, we appreciate Reviewer 3’s insightful suggestion and consider it an important direction for future studies with larger cohorts.

Reviewer 3:  In the Figure 4. (A), please make sure that c is preoperative MRI or postoperative MRI?

Our answer: We confirm to Reviewer 4 that the MRI images in Figure 4 (A: a-b-c) correspond to the preoperative MRI. However, in the updated version of the article, and in accordance with Reviewer 2's suggestions, Figure 4 of the original article has been removed

Reviewer 3: This paragraph of Pathophysiology of BEH could be shortened and placed at the beginning of the discussion section.

Our answer: Based on the suggestions from Reviewer 3, the section discussing the pathophysiology of BEH has been relocated to the beginning of the discussion section. Additionally, the overall discussion has been significantly shortened, except for the segment on ICP findings in children with BEH, as this is the focus of the article.
